# Material Characterization, Thermal Analysis, and Mechanical Performance of a Laser-Polished Ti Alloy Prepared by Selective Laser Melting

**Yu-Hang Li** [1,†], **Bing Wang** [2,†], **Cheng-Peng Ma** [1], **Zhi-Hao Fang** [3], **Long-Fei Chen** [3,*], **Ying-Chun Guan** [1,4,5,*] and **Shou-Feng Yang** [6]

[1] School of Mechanical Engineering and Automation, Beihang University, 37 Xueyuan Road, Beijing 100191, China; lyhliyuhang@buaa.edu.cn (Y.-H.L.); macp1107@buaa.edu.cn (C.-P.M.)
[2] ST Engineering—NTU Corporate Laboratory, School of Electrical and Electronic Engineering, Nanyang Technological University, 61 Nanyang Drive, Singapore 637335, Singapore; wangbing@ntu.edu.sg
[3] School of Energy and Power Engineering, Beihang University, 37 Xueyuan Road, Beijing 100191, China; zy1604604@buaa.edu.cn
[4] National Engineering Laboratory of Additive Manufacturing for Large Metallic Components, Beihang University, 37 Xueyuan Road, Beijing 100191, China
[5] Hefei Innovation Research Institute of Beihang University, Xinzhan Hi-tech District, Hefei 230013, China
[6] Additive Manufacturing Research Group, KU Leuven, Celestijnenlaan, 300- Box 2420, 3001 Leuven, Belgium; shoufeng.yang@kuleuven.be
* Correspondence: chenlongfei@buaa.edu.cn (L.-F.C.); guanyingchun@buaa.edu.cn (Y.-C.G.);
  Tel.: +86-10-8233-9691 (L.-F.C.); +86-10-8231-7430 (Y.-C.G.)
† Yu-Hang Li and Bing Wang contributed equally to this work.

**Abstract:** The laser polishing technique offers an adaptable, accurate, and environmentally friendly solution to enhance the surface quality of additive manufactured metallic components. Recent work has shown that the surface roughness of laser additive manufactured metallic alloys can be significantly reduced via the laser polishing method. This paper examines the mechanical performances of a laser polished surface fabricated by selective laser melting (SLM). Compared with the original SLM surface, systematic measurements revealed that the surface roughness of the laser polished surface can be effectively reduced from 6.53 μm to 0.32 μm, while the microhardness and wear resistance increased by 25% and 39%, respectively. Through a thermal history analysis of the laser polishing process using the finite element model, new martensitic phase formation in the laser polished layer is carefully explained, which reveals significant effects on residual stress, strength, and fatigue. These findings establish foundational data to predict the mechanical performance of laser polished metallic components fabricated by additive manufacturing methods, and pave the way for functional surface design with practical application via the laser process.

**Keywords:** laser polishing; additive manufacturing; selective laser melting; thermal kinetics; mechanical properties

---

## 1. Introduction

The Ti-6A-4V alloy has been widely employed in the aerospace and medical device industry due to its high specific strength, corrosion resistance, and biocompatibility [1]. In recent years, selective laser melting (SLM) has attracted much attention as a promising method to fabricate complex geometries, as well as new microstructures with novel properties, including an excellent mechanical performance in terms of yield, tensile strength, and ductility [2,3]. However, it was reported that

the surface quality of the SLMed surface was poor due to the uneven distribution of powders at the surface caused by the waviness of laser scan tracks and the layered geometries [4–6], and rough surfaces generally limit the practical applications of the entire component. Among the most desirable properties of SLM components, a high surface finish and good surface performance are crucial for the direct mechanical application of SLM components in the future. Conventional post processes of metallic additive-manufactured components include manual polishing, chemical polishing, electrolytic polishing, mechanical milling, ultrasonic polishing, and so on [7]. Unfortunately, difficult to machine materials, geometrical flexibility, environmental pollution, a low processing efficiency, and being incompatible with the automation of these methods limit their application and development [7].

The laser polishing technique has been developed as an effective method to polish metal surfaces, during which process the molten material is redistributed and solidified under surface tension, and gravity is caused during the local melting and cooling process [8]. Recent advances in laser polishing have established the foundations for the physical mechanism responsible for topography evolution of the metallic additive manufactured surface. Lamikiz et al. carried out laser polishing on a bronze alloy made by selective laser sintering (SLS), and found that the laser polished layer showed a homogeneous composition and high microhardness with the reduction of surface roughness from 7.5 μm to 1.49 μm [4]. Ukar et al. presented a methodology to predict surface topography on laser polished surfaces with good agreement in experimental validation on DIN 1.2379 tool steel and reduced the surface roughness from Ra 7.5 μm to 1.49 μm [9]. Marimuthu et al. investigated the surface polishing of titanium (Ti) components fabricated by SLM with the surface roughness reduced from 10.2 μm to 2.4 μm [10]. Wang et al. built a surface prediction model for thermocapillary regime pulsed laser micro polishing (PLμP) and the model was verified by area polishing of the titanium alloy Ti-6Al-4V and S7 tool steel and the predicted average surface roughnesses were within 15% of the measured values [11]. Rosa et al. examined the effect of laser polishing on stainless steel and Ti surfaces via laser metal deposition (LMD), and provided statistical models to determine the settings of operating parameters according to surface roughness specification. The results obtained a final surface roughness of 0.79–0.23 μm for an initial Sa of 21–1.8 μm [12]. Bhaduri et al. reported laser polishing results in AM 316L stainless steel with the reduction of surface roughness from 3.8 μm to 0.2 μm [13]. In our recent work, we discussed the effect of laser polishing on the mechanical properties of LMD Ti alloys, where the surface roughness of LMD Ti alloys was reduced from 5 μm to 1 μm, and the microhardness and wear resistance of the polished layer were enhanced [14]. However, these is little comprehensive analysis of the thermal cycle, microstructure, surface mechanical properties, and relationship between them regarding the laser polished layer.

In the present work, the effects of laser polishing on the mechanical properties and surface topography of a Ti alloy manufactured by SLM are discussed. Besides, finite element simulation is used to investigate the temperature field and historical cycle during laser polishing. An attempt has been made to establish the relationship between the thermal history and microstructure evolution of the laser-polished surface. Systematic measurements have been examined, with an emphasis on underlying mechanistic reasoning including microhardness, wear resistance, residual stress, tensile properties, and high cycle fatigue life.

## 2. Materials and Methods

The SLM Ti-6Al-4V (wt.%) alloy material was annealed at 1003 K (730 °C) for 2 h and then slow cooled in a vacuum furnace to 803 K (530 °C). After that, the material was cooled to room temperature in Argon atmosphere via a switched off furnace and air cooling before the investigation of laser polishing. The specimens were polished using a wavelength of 1060 nm continuous fiber laser (Shengxiong Laser, Shenzhen, China) in a high purity Ar gas environment. Optimal laser polishing process parameters were explored, as shown in Table 1. The specimens were numbered as shown in the first column of Table 1, and were also presented with the scanning velocity. For example,

specimen 1–150 means a specimen processed with 70 W (10% overlapping ratio) and with a 150 mm/s scanning velocity.

**Table 1.** Laser polishing parameters of different samples.

| Specimen | Laser Power (W) | Scanning Velocity (mm/s) | Overlapping Ratio (%) |
|----------|-----------------|--------------------------|-----------------------|
| 1-X | 70 | 50; 100; 150; 200; 250 | 10 |
| 2-X | 80 | 50; 100; 150; 200; 250 | 10 |
| 3-X | 90 | 50; 100; 150; 200; 250 | 10 |
| 4-X | 100 | 50; 100; 150; 200; 250 | 10 |

The comprehensive use of optical microscopy (OM, LV150N, Nikon, Tokyo, Japan) and scanning electron microscopy (Hitachi SU8010, Zeiss, Tokyo, Japan) was employed to observe the surface topography. The surface profile was measured by a Keyence VK-X series laser confocal microscope (VK100, Keyence, Osaka, Japan). An X-ray diffractometer (Rigaku D/max2200PC, SHIMADZU, Tsushima, Japan) was used to debate the phase evolution in the laser-polished layer after laser polishing.

Microhardness was measured by Vickers (Shaoxing jingbo, Shaoxing, China) with the loading force of 1.96 N and six measurements were averaged as the microhardness of each data point. A dry sliding friction test was carried out using the CETR UMT-2 friction testing system ($Al_2O_3$ balls of 5 mm as the counterpart) with a 50 N load at a sliding velocity of 200 mm/min. The residual stress of laser polished and as-received samples was measured by the XRD method (XA-350X) with a voltage of 28 KV and current of 8 mA, and the stress results were obtained directly from integrated software (Proto iXRD® X-ray, Siemens, Hanover, Germany).

Tensile properties were measured using the SANS tensile testing machine (Wen Teng, Jinan, China) at room temperature. Based on GB/T 228.1, tensile specimens with a thickness of 3 mm and gauge length of 39 mm were milled. The Instron 8801 fatigue testing system was used for high cycle fatigue (HCF) tests at two stress levels of 500 and 600 MPa, the selected mean stress condition was R = 0.1 for the stress ratio, and the test frequency was 10 Hz. According to GBT 3075-2008, the cylindric specimens were used in these tests and the diameter of the test section was 3.0 mm. The surfaces of tensile and fatigue samples were completely polished by the laser before testing.

The temperature field and thermal cycle during the laser polishing process have been calculated by the finite element simulation method. The commercial software ABAQUS (ABAQUS 6.13, SIMULIA, Rhode Island, RI, USA) has been used to simulate the temperature field at the surface of the SLMed Ti alloy during laser polishing. The thermal model was established by solving the general heat transfer equation for conduction, and evaluating the heat flow in each differential volume element and differential time [15]. In our model, the calculation has been conducted on a rectangle substrate with dimensions of $4 \times 2 \times 0.5$ mm, and the minimum differential volume element is a cube with a size of $30 \times 30 \times 30$ μm. The density of the input heat flow rate of each facula is calculated by the Gaussian function and it is consistent with the laser heat source used in our laser polishing experiment.

## 3. Results and Discussions

### 3.1. Surface Topography

Figure 1 displays the results of as-received and laser polishing SLMed Ti alloy surface topography. Figure 1a shows that the original SLM surfaces and surface roughness values Ra and Rz were 6.53 μm and 140.29 μm, respectively, which were attributed to the large number of partially melted powders that accumulated at the surface during the SLM process [6]. Figure 1b shows the roughness results of samples listed in Table 1 and the best laser polishing quality is obtained when the scanning velocity is 150 mm/s with a 10% overlapping ratio and 90 W laser power. Significant improvement of surface finishing is observed, as shown in Figure 1c,d. After laser polishing, surface roughness Ra and Rz values were reduced to 0.32 μm and 6.42 μm, respectively. The measurement results of surface

topography indicate that the partially melted powders were further melted at the laser-polished surface and hence the surface appeared smooth.

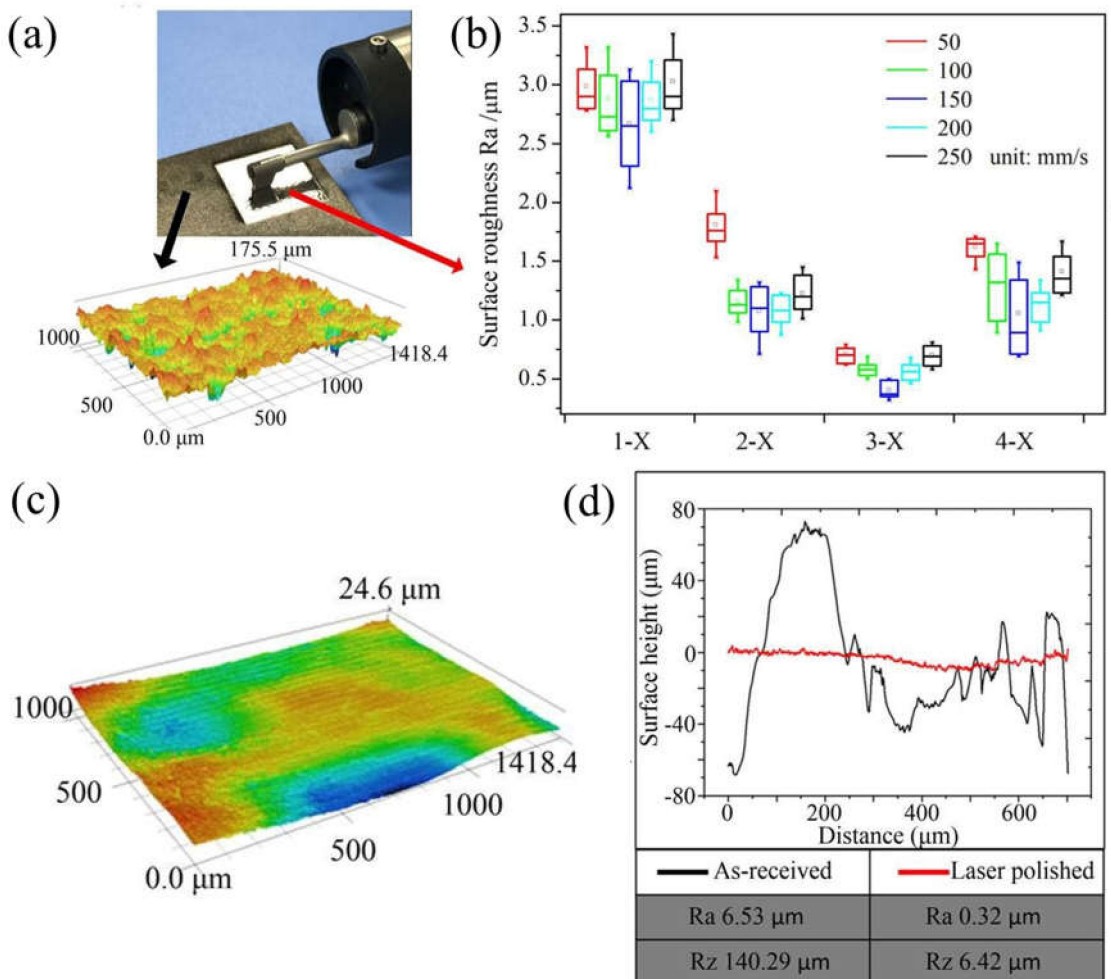

**Figure 1.** Effects of laser polishing on the SLM Ti alloy surface: (**a**) The topography of the as-received surface; (**b**) the roughness test results of the twenty specimens; (**c**) the optimized topography of the laser polished surface; and (**d**) the profiles of both as-received and optimized laser polished regions.

According to previous findings, the Marangoni flow was the main factor driving the liquid flow and it was produced by a non-uniform temperature or concentration distribution on the interface of two fluids [16]. The surface tension at the cooler edge is higher than the molten pool centre due to the negative temperature gradient of current surface tension [17]. Accordingly, Marangoni flow that is radially outward results in a depression of the liquid surface under the laser beam and ridging of the liquid surface elsewhere. When the laser beam moves to the next area of the irradiated surface, deformation of the liquid surface caused by thermally-driven Marangoni flow and the molten material were redistributed by surface tension and gravity. Finally, a smoothened surface was realized through laser polishing.

*3.2. Microstructure Evolution*

A three-dimensional numerical fluid flow and heat transfer model was built to investigate the rapid melting and solidification caused by laser processing, and the exhaustive explanation of its convergence criteria, boundary conditions, and governing equations was provided in previous work [17]. In our research, the temperature distribution curve shown in Figure 2b was calculated based

on the transient heat balance equation in the model. Moreover, the surface tension was identified by the surface-tension Reynolds number ($M_a$) as Equation (1) [16].

$$M_a = \frac{\rho L_R \Delta T |\partial\gamma/\partial T|}{\mu^2} \tag{1}$$

where $\Delta T$ is the temperature difference between the solidus temperature and peak pool temperature. He et al. provided more detailed information about this equation [17]. In the molten zone, peak temperature reduced from 2800 °C to 1600 °C from the top surface to the heat affected zone (HAZ). Namely, $\Delta T$ in the molten zone significantly reduced. In addition, Guan et al. explained that Marangoni flow apparently increases when $\Delta T$ is large on the basis of the Reynolds number [16]. Because of the rapid solidification rate and huge temperature gradient associated with Marangoni flow, free surface oscillates much more strongly near the solidification front; thereby, the roughness of the solidified surface significantly reduced, as exhibited in Figure 1.

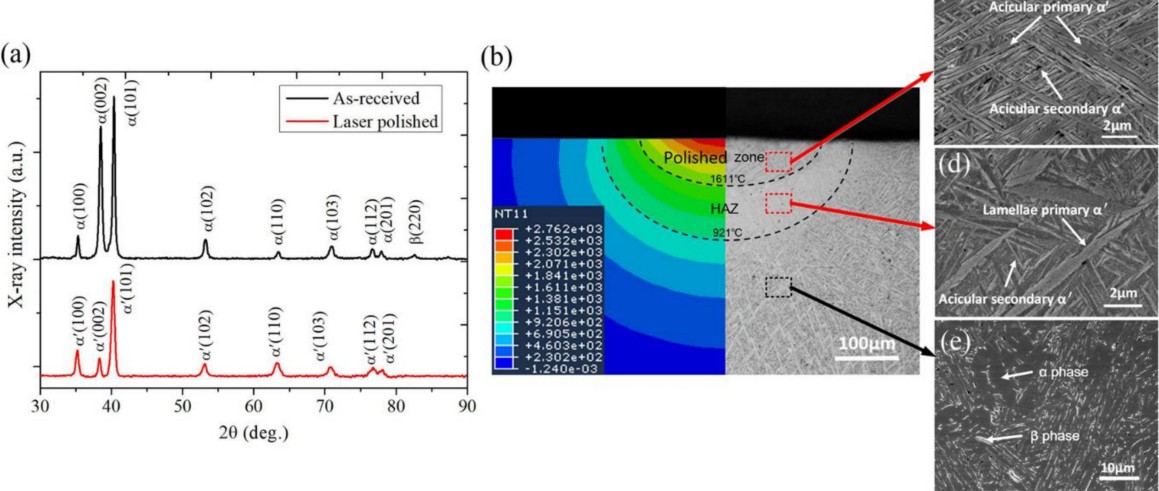

**Figure 2.** The XRD analysis, microstructure, and finite element simulation: (**a**) XRD patterns of as-received and laser polished SLM Ti alloy surfaces; (**b**) temperature distribution of cross-section view, with microstructure of the polished zone (**c**), HAZ (**d**), and as-received zone (**e**).

It is known that the microstructure of the SLMed Ti alloy contains needle-shape β phase distributed non-homogeneously in an α matrix phase [18,19], as shown in Figure 2e. The cross-section view of the laser polished surface was further investigated in Figure 2b, where the melted depth is nearly 60 μm and the depth of HAZ is ordinarily less than 100 μm. The solidification microstructure in the molten zone presented the primary acicular α′ phase and fine secondary α′ phase precipitated at dendrite/cellular boundaries, as shown in Figure 2c. The XRD analyses in Figure 2a reveals that the laser polished layer was made up of martensite α′ phase. The acicular primary structures are the main morphology of the solidification microstructure owing to liquid-solid interface stability factor G (temperature gradient)/R (solidification rate) for martensitic transformation [20]. According to previous work, the reason why the acicular primary structures formed is the rejection of solute atoms in the normal direction to the average solid/liquid interface from its tip and lateral surface on the basis of cellular structures [17]. Moreover, the refined microstructure is not uniform in the laser polished layer and the formation of secondary α′ phase might be due to the inadequate energy distribution on the edge of the Gauss beam. At the bottom of the laser polished layer, a coarse lamellae α′ structure was found as a result of the "quench-hold-quench" process caused by laser re-heating cycles during the scanning process. During coarsening, small precipitates shrunk and eventually vanished together, whilst large precipitates grew at their expense by solid-state diffusion. This resulted in a coarse lamellae structure in the microstructure, as shown in Figure 2d. The dimension of the

microstructure with an average size of 9.8 µm in the laser-polished layer is smaller than that of the substrate with an average size of 48.6 µm due to the rapid re-melting and solidification process [21].

### 3.3. Thermal Analysis

The formation of $\alpha'$ martensite in both the polished layer and HAZ could be further explained by thermal cycle analysis, as shown in Figure 3. The schematic diagram of the laser scanning path and temperature distribution is shown in Figure 3a. The calculated Ti-V binary phase diagram [22] and continuous cooling diagram of Ti6Al4V [23] are shown in Figure 3b,c, respectively.

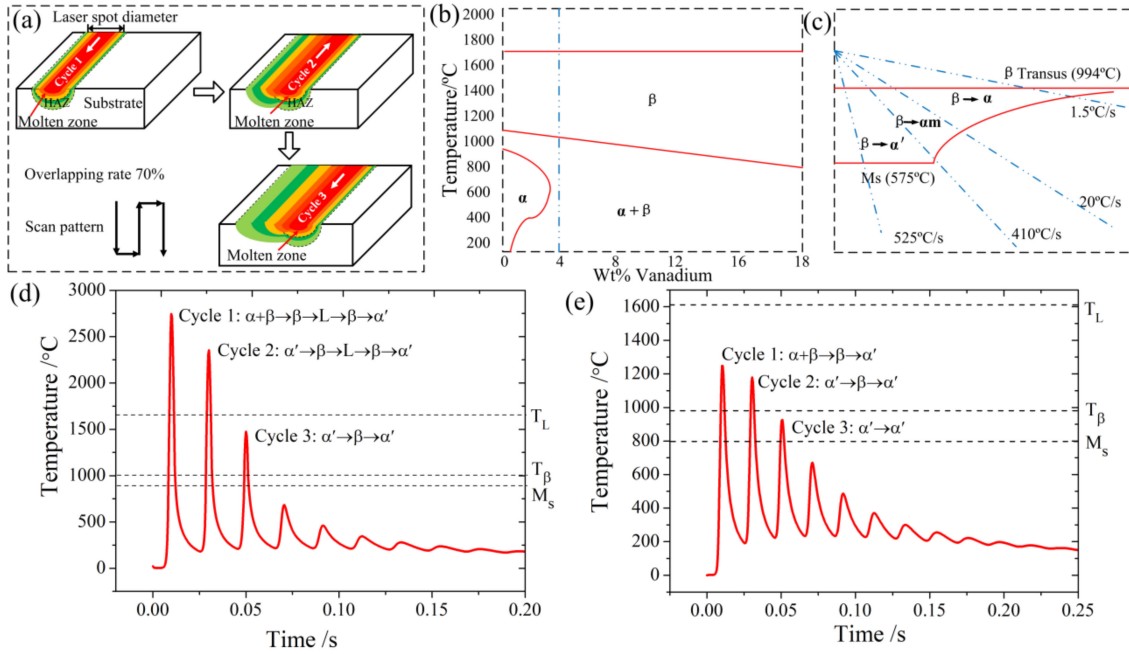

**Figure 3.** Thermal analysis of laser polishing process: (**a**) Laser scanning pattern and temperature distribution; (**b**) Ti-V binary phase diagram [22]; (**c**) continuous cooling diagram for Ti-6Al-4V [23]; (**d**) thermal cycle in melted layer; (**e**) thermal cycle in HAZ.

In the molted zone of the laser-polished layer, two peak temperatures of cycle 1 and 2 were in excess of liquidus temperature $T$L, as displayed in Figure 3a. Solidification transformation of L → β → $\alpha'$ (acicular primary martensites) [24] was observed when the molten material cooled to temperatures lower than Ms (martensite transformation start temperature as indicated in phase diagram Figure 3d) and the cooling rate was calculated to be higher than 106 °C/s. After thermal cycle 3 produced by the third laser scanning, the peak temperature was in the range of β transus temperature $T$β to $T$L. Thereby, all previous $\alpha'$ martensites transformed to β phase under solid state phase transformations. According to previous study, it was proposed that some dislocations may remain within β phase, which would promote the nucleation and formation of primary and secondary $\alpha'$ martensites during the cooling process [18]. In HAZ, two peak temperatures of cycle 1 and 2 were higher than $T$β, but lower than $T$L, in Figure 3e, correspondingly, and solid state phase transformations were similar for thermal cycle 3 of the melted layer, leading to β phase being transformed into $\alpha'$ martensites. In the case of cycle 3, primary $\alpha'$ martensite underwent coarsening, and may have partially decomposed into metastable β phase due to the peak temperature in the range of $M$s to $T$β. Therefore, metastable β phase could transform into acicular $\alpha'$ martensites in the cooling process. The HAZ microstructure of SLM Ti-6Al-4V was made up of coarse primary and acicular secondary $\alpha'$ martensite and our result is in agreement with Yang's findings [22,25].

### 3.4. Mechanical Properties

The microhardness distribution in the laser-polished layer is shown in Figure 4. The average microhardness of the substrate is about 340 HV, while the microhardness of the polished layer increased to about 426 HV, which is approximately 25% higher than that of substrate material. Moreover, Figure 5 shows the cross-section profiles of wear scars of both the original SLM and laser polished surface. The EDS result of the before and after laser polishing area is shown in Figure 6, and it can be seen that the oxygen content of the laser polished surface did not increase with argon gas protection. The wear scar on the laser polished sample was shallower than on the original SLM sample. The wear rate can be calculated for quantitative comparison of the samples' wear resistance [19], and it is defined as Equation (2).

$$W = \frac{\Delta V}{L \times D} \tag{2}$$

where $\Delta V$ is the volume of the worn material, $D$ is the sliding distance, and $L$ is the load used in the test. Therefore, the wear rate of the laser polished surface is 39% lower than the original SLM surface, indicating that laser polishing enhanced the wear resistance of the SLM surface. Based on the Archard's wear equation, the wear volume is inversely proportional to the hardness of the studied alloy [26,27]. The improvement of microhardness and wear behavior is mainly attributed to the emergence of $\alpha'$ martensitic phase and the fine grain strengthening in the laser polished layer without considering changes in oxygen content, because $\alpha'$ martensitic has a hexagonal closed-packed (hcp) structure with a higher bulk modulus value than that of $\alpha$ and $\beta$ phase [28–30].

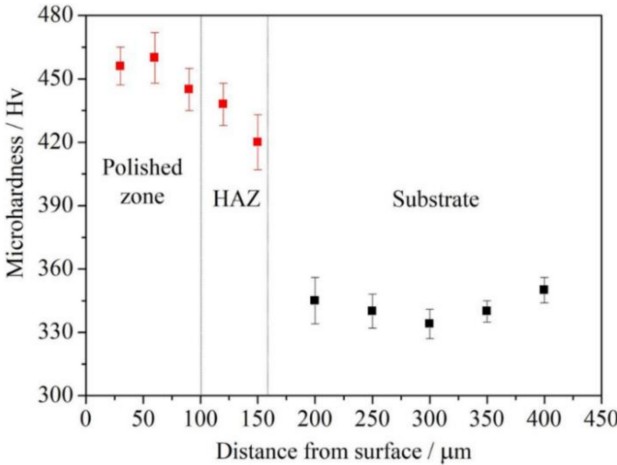

**Figure 4.** Microhardness distributions in the laser-polished layer.

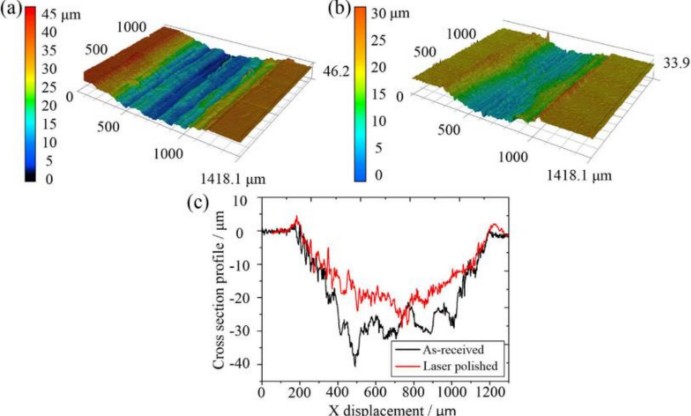

**Figure 5.** Cross-section profile of wear scars: (**a**) As-received surface; (**b**) laser polished surface; (**c**) cross-section profile.

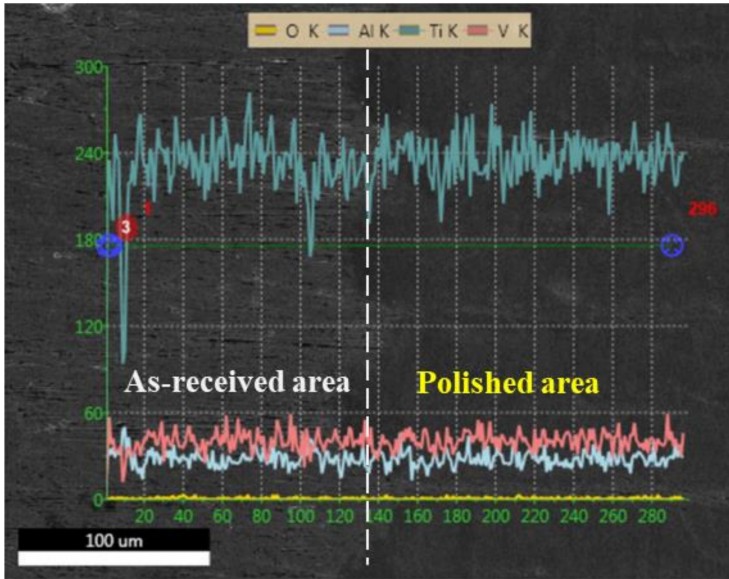

**Figure 6.** The EDS result of as-received and polished areas.

Figure 7 shows the residual stress of the original and laser polished surfaces. Figure 7a illustrates two scanning modes of laser polishing which are parallel to the building direction (BD) and scanning direction during the SLM process. The result of residual stress is displayed in Figure 7b, and there is no residual stress on the as-received surface and the residual stress is in the range of 300 MPa to 500 MPa on the laser polished surface by two scanning modes. It is known that laser polishing produces residual tensile stress on the polished surface due to the rapid cooling and solidification [22], which is irrelevant to the direction of laser scanning during laser polishing.

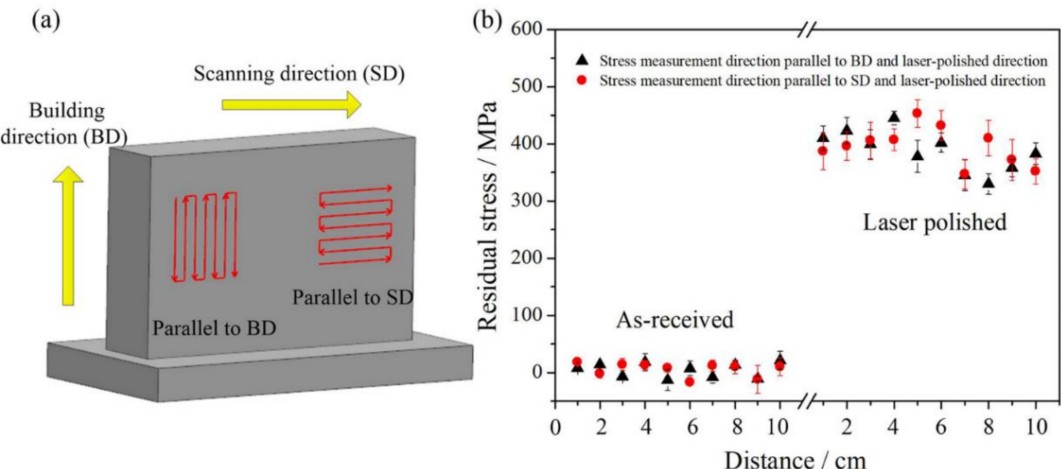

**Figure 7.** Residual stress analysis: (**a**) Laser polished direction; (**b**) residual stress of as-received and laser polished surfaces.

Tensile behaviors, including the ultimate tensile strength, yield strength, and elongation of as-received and laser polished samples, are shown in Figure 8a. Relative to as-received SLM samples, the variation of tensile strength and yield strength of laser polished specimens was less than 1%, while elongation was decreased by 5%. Due to the thickness of tensile samples being 3 mm, the laser polishing layer was only about 60 μm. Therefore, there was no significant change in the tensile properties after laser polishing. High cycle fatigue life of as-received and laser-polished specimens is displayed in Figure 8b. Most of the as-received specimens reached $10^7$ cycles, while the fatigue life of laser polished specimens was reduced to the range of $10^4$ to $10^5$ cycles. Figure 8d exhibits a fatigue

fracture surface of the laser-polished specimen, and the fatigue source was located at the polished layer, as shown in Figure 8e. The magnified view of over loading area showed ductile fracture with shallow dimples in Figure 8c, which demonstrates the good ductility of substrate material. Vilaro et al. [19] investigated the impact of the microstructure on mechanical properties using the SLMed Ti-6Al-4V alloy, and found that the formation of α′ martensitic phase raised the tensile strength from 1.13 to 1.35 GPa and decreased elongation from 8% to 3% compared to that of α and β phase in the substrate. Accordingly, we propose that the change in fatigue properties after laser polishing is possibly attributed to the combined effect of SLM defects, including pores or micro-cracks, as well as precipitation of brittle α′ martensitic, which weaken the fatigue properties of laser polished specimens. Besides, we also consider that the residual tensile stress may weaken the fatigue properties, which will accelerate the fatigue crack propagation of the laser polishing layer. Therefore, future work should focus on reducing defects of SLM and the formation of brittle α′ martensitic phase, as well as suppressing the formation of residual tensile stress.

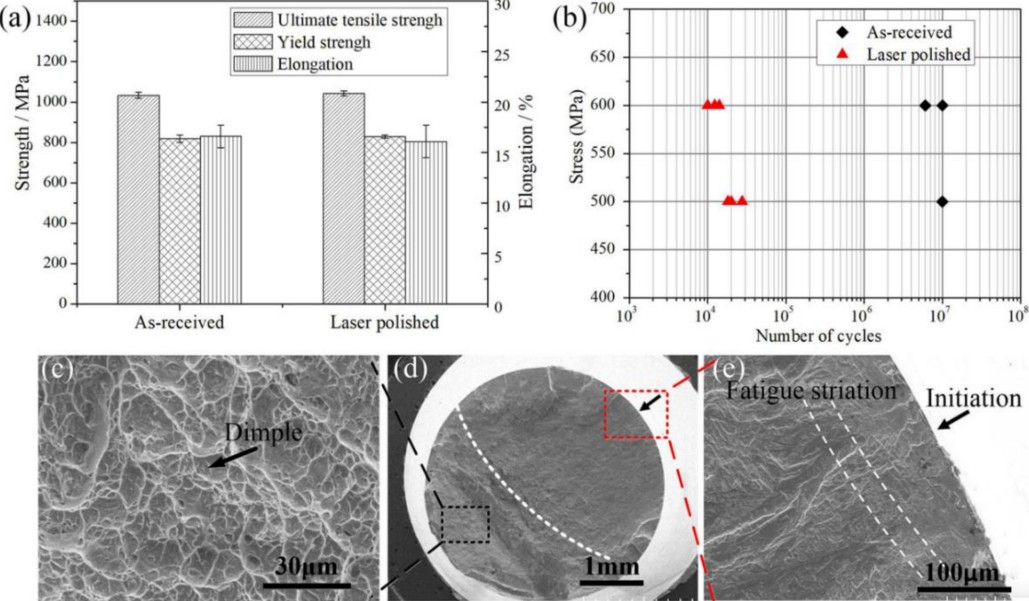

**Figure 8.** (**a**) Tensile properties; (**b**) high cycle fatigue life; (**c**) dimples due to overload in (**d**); (**d**) overview of fracture surface; (**e**) crack initiation area at the laser polished surface in (**d**).

## 4. Conclusions

The surface topography, solidification microstructure, and mechanical performance of the SLMed Ti alloy after laser polishing have been investigated. Surface roughness Ra of initial SLM specimens was reduced from 6.53 μm to 0.32 μm after laser polishing. For the laser polished surface, both microhardness and wear resistance were enhanced by 25% and 39%, respectively, relative to the as-received surface. Laser polishing resulted in residual tensile stress on the material surface and the residual stress was independent of laser scanning direction. Compared with the as-received SLM surface, the tensile strength and yield strength of laser polished surface were not much affected, while elongation performance slightly dropped. The high cycle fatigue life of the laser-polished surface was reduced for the formation of the martensitic layer with poor ductility. It was found that the changes in mechanical properties were intrinsically related to the formation of new α′ martensitic phase by thermal analysis of the laser polishing process.

**Author Contributions:** Y.-H.L. and B.W. conducted the original writing and experimental work; Z.-H.F. and C.-P.M. conducted data analysis; L.-F.C. conducted the review and editing of the manuscript; and Y.-C.G. had a role in supervision, project administration, and funding acquisition, and S.-F.Y. has provided valuable suggestions of the revision.

**Funding:** This work was supported by the National Key Research and Development Program of China under Grant 2018YFB1107400, 2018YFB1107700, and 2016YFB1102503; National Key Basic Research Program of China under Grant 2015CB059900; National Natural Science Foundation of China under Grant 51705013; and Beijing Natural Science Foundation under Grant J170002 and 3162019.

**Conflicts of Interest:** The authors declare no conflict of interest.

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
