# Peer review of "Material Characterization, Thermal Analysis, and Mechanical Performance of a Laser-Polished Ti Alloy Prepared by Selective Laser Melting"

_metals, doi:10.3390/met9020112_

Reviewer 1 Report

This paper reports an investigation of the change by laser polishing of the surface morphology, crystal structure and mechanical properties including residual stress, strength and fatigue of titanium alloys fabricated by selective laser melting.. Significant reduction of the surface roughness of the materials, retainment of the crystal structure of the original material, increase of the microhardness and of residual stress after laser polishing, are reported in the paper and they are attributed to the formation of a new α' martensitic phase determined by thermal analysis of the laser polishing process. The work is original, well performed and reported, paper well written thus publication of the paper is recommended.

Author Response

Thank you for your comments.

Reviewer 2 Report

This manuscript presents investigations characterizing the surface of a laser polished Ti6Al4V fabricated via selective laser melting. Although the material response to this process is relevant in additive manufacturing, the manuscript’s structure is poor, issues remain unclear, and the following issues must be considered:

Title: Thermal Kinetics and Mechanical Performance of Laser-Polished Ti Alloy Fabricated by Selective Laser Melting.  

The thermal analysis performed are associated with the suface while mechanical performance with the entire bulk alloy.  Thus, the title is misleading and must be changed for better understanding.

Thermal Kinetics is an inadequate terminology since the kinetics has not been experimentally measured, for instance, using DSC or dilatometry.

Were the cylindric tensile samples completely “lasered” prior testing? It is not explained.  

The introduction is basically focused on surface modification by laser polishing. Several  previous works are mentioned without explaining the novelty of the present work. Also, it is mentioned that „The current trend with metallic additive manufacturing (AM) lies in material and structure design“ without clarifying what does it mean for the reader. Further, concise explanation must include the development of new alloys and new SLM strategies: e.g. see  Peritectic titanium alloys for 3D printing, Nat. Commun., 9, 3426 (2018),  Inducing stable alpha+beta microstructures during selective laser melting of Ti-6Al-4V using intensified intrinsic heat treatments, Materials 10,268 (2017).

Figure 1 accumulates considerable information, though its items are too small: difficult to read. The information must be clearly presented, and big enough.

The annealing treatment of the initial ti64 condition must be explained.

Line 147: the matrix is not alpha, it is beta. Also, SLMed Ti64 is not usually as shown in fig 2e (the initial condition studied here –SLMed and post annealed), it is a martensitic condition similar to fig. 2c. Which annealing is used to get the as-received condition?  

Line 154: “The dendrite structures are the main morphology of solidification microstructure…”.  No dendrite structures are presented in the results. Please, clarify it.

Lines 157-159- this is an speculation. If so, the formation  mechanism of secondary martensite plates must be explained. In which conditions do they form etc.?

Lines 160-161: the martensite plates in fig.2d are formed athermally and thus, cannot increase their thickness by diffusion. Otherwise, martensite decomposition occurs and alpha +beta is abtained.this is wrong.

Section 3.3. does not present own results. It correspond to a compendium of literature to explain the possible transformations during lasering. This has already been widely discussed in the literature. That the martensite transforms to martensite again upon cooling from Ms-beta it is an speculation. The phase diagram is considered, though SLM kinetics are more metastable. Experimental in situ investigations not mentioned exist in this regard.

The methodology used to calculate residual stresses is not fully described (which Do, phases and peaks were taken etc).    

The polished and no polished samples show the same performance during tensile testing. No changes are observed.

English grammar requires revision throughout the manuscript in order to improve readibility.

Author Response

We would like to thank the Reviewers’ invaluable comments. We have carefully considered all the comments and made corresponding modifications in the revised manuscript accordingly as listed below. All the modifications has been done are highlighted with in BLUE colour in the revised manuscript.

Note: In the authors’ reply to the reviewers, all paper numbers, figure numbers, and references are refer to the revised manuscript.

Reviewers' comments:

Reviewer #2:

This manuscript presents investigations characterizing the surface of a laser polished Ti6Al4V fabricated via selective laser melting. Although the material response to this process is relevant in additive manufacturing, the manuscript’s structure is poor, issues remain unclear, and the following issues must be considered:

Title: Thermal Kinetics and Mechanical Performance of Laser-Polished Ti Alloy Fabricated by Selective Laser Melting. 

1.     The thermal analysis performed are associated with the surface while mechanical performance with the entire bulk alloy. Thus, the title is misleading and must be changed for better understanding.

Thermal Kinetics is an inadequate terminology since the kinetics has not been experimentally measured, for instance, using DSC or dilatometry.

Reply: Thank you for your advice. We have changed the title as “Material Characterization, Thermal Analysis and Mechanical Performance of Laser-polished Ti Alloy Prepared by Selective Laser Melting”.

2. Were the cylindric tensile samples completely “lasered” prior testing? It is not explained. 

Reply: The surface of cylindric tensile samples has been completely polished by laser before testing, and the detailed information has been added in Line 106-107 on Page 3.

3.   The introduction is basically focused on surface modification by laser polishing. Several previous works are mentioned without explaining the novelty of the present work. Also, it is mentioned that ”The current trend with metallic additive manufacturing (AM) lies in material and structure design” without clarifying what does it mean for the reader. Further, concise explanation must include the development of new alloys and new SLM strategies: e.g. see  Peritectic titanium alloys for 3D printing, Nat. Commun., 9, 3426 (2018),  Inducing stable alpha+beta microstructures during selective laser melting of Ti-6Al-4V using intensified intrinsic heat treatments, Materials 10,268 (2017).

Reply: Thank you for your advice. We have carefully modified the Introduction, and the novelty of the present work has been explained. The detailed information has been added in Line 60-61, Line 64-65 and Line 68-69 on Page 2. Besides, we have revised the sentence “The current trend with metallic additive manufacturing (AM) lies in material and structure design”, and the detailed information has been added in Line 38-42 on Page 1. Moreover, the important application of titanium alloys and the problems of AM surface quality has been highlighted on basis of the given references.

4.   Figure 1 accumulates considerable information, though its items are too small: difficult to read. The information must be clearly presented, and big enough.

Reply: We have carefully revised the Figure 1. The detailed information has been added on Page 4.

5.   The annealing treatment of the initial ti64 condition must be explained.

Reply: The Ti-6Al-4V (wt. %) titanium alloy was annealed at 1003 K (730 ℃) in this paper. The detailed information has been added in Line 84-85 on Page 2.

6.   Line 147: the matrix is not alpha, it is beta. Also, SLMed Ti64 is not usually as shown in fig 2e (the initial condition studied here –SLMed and post annealed), it is a martensitic condition similar to fig. 2c. Which annealing is used to get the as-received condition? 

Reply: Thank you for your advice. The SLM specimen was annealed at 1003 K (730 ℃), therefore, needle shape β phase was distributed non-homogeneously among α phase as shown in Fig.2(e).

7.   Line 154: “The dendrite structures are the main morphology of solidification microstructure…”.  No dendrite structures are presented in the results. Please, clarify it.

Reply: We have carefully examined the sentence, and we revised the sentence “The acicular primary structures are the main morphology of solidification microstructure…” according to your advice. The detailed information has been revised in Line 168-170 on Page 5. Similar results can be seen in the reference “Xu, W.; Lui, E. W.; Pateras, A.; Qian, M.; Brandt, M., In situ tailoring microstructure in additively manufactured Ti-6Al-4V for superior mechanical performance. Acta Mater 2017, 125, 390-400.” and “Ma, C. P.; Guan, Y. C.; Zhou, W., Laser polishing of additive manufactured Ti alloys. Opt Laser Eng 2017, 93, 171-177.”

8.   Lines 157-159- this is an speculation. If so, the formation mechanism of secondary martensite plates must be explained. In which conditions do they form etc.?

Reply:

According to the findings of Reference as “Barriobero-Vila, P.; Gussone, J.; Stark, A.; Schell, N.; Haubrich, J.; Requena, G., Peritectic titanium alloys for 3D printing. Nature communications 2018, 9, (1), 3426. ” And “Ahmed, T. & Rack, H. J. Phase transformations during cooling in α+β titanium alloys. Mater. Sci. Eng. A 243, 206–211 (1998)”, the formation of α′ martensite in Ti-6Al-4V alloy occurs when cooling rate is above 410 °C·s-1. In this work, cooling rate of laser-polished layer has been calculated as 1.209×104 K/s on basis of heat-flow model. Therefore, we claim that β phase transforms completely into metastable α′ martensitic phase by diffusionless transformation of parent β grains (primary high temperature phase) in the as-received of α + β Ti alloys. Moreover, the results has been justified by the results of XRD and SEM investigation, which is in agreement with the reference “Ma, C. P.; Guan, Y. C.; Zhou, W., Laser polishing of additive manufactured Ti alloys. Opt Laser Eng 2017, 93, 171-177.”

9.   Lines 160-161: the martensite plates in fig.2d are formed athermally and thus, cannot increase their thickness by diffusion. Otherwise, martensite decomposition occurs and alpha +beta is abtained. this is wrong.

Reply: Laser polishing is based on thermal processing with rapid re-melting and cooling. We confirm that the laser-polished layer consists of martensite α’ phase according to the results of XRD and SEM investigation as well as thermal analysis, while the as-received layer consists of alpha +beta phase. These phase structure can be seen in Fig.2 (d) and the Fig.2 (e).

10. Section 3.3. does not present own results. It correspond to a compendium of literature to explain the possible transformations during lasering. This has already been widely discussed in the literature. That the martensite transforms to martensite again upon cooling from Ms-beta it is an speculation. The phase diagram is considered, though SLM kinetics are more metastable. Experimental in situ investigations not mentioned exist in this regard.

Reply: Thank you for your advice. It should be noted that the Fig.3 (b) and (c) in section 3.3 is based on phase diagram from the literature, however, Fig. 3 (a), (d) and (e) has been obtained on basis of our finite element simulation results. Due to high temperature and complex phenomenon during laser polishing, it is difficult to measure in situ temperature of molten pool directly by experimental methods. Therefore, model-based thermal simulation with experimental verification method has been widely employed by researchers, such as “Ukar, E., Lamikiz, A., Martínez, S., Tabernero, I., Lacalle, L.N.L.d., 2012. Roughness prediction on laser polished surfaces. J Mater Process Tech 212, 1305-1313.” and “S. Marimuthu, A. Triantaphyllou, M. Antar, D. Wimpenny, H. Morton, M. Beard, Laser polishing of selective laser melted components, Int. J. Mach. Tools Manuf. 95 (2015) 97-104.”

In this work, the microstructures were further analysed and verified with experimental results in 3.2 section. After laser polishing, laser-polished layer mainly consists of α’ martensitic phase because of the phase transformation α+β→β→α’. Similar results can be verified in the reference“Ma, C. P.; Guan, Y. C.; Zhou, W., Laser polishing of additive manufactured Ti alloys. Opt Laser Eng 2017, 93, 171-177.”

11. The methodology used to calculate residual stresses is not fully described (which Do, phases and peaks were taken etc).   

Reply: The residual stress was measured by XRD method (XA-350X) with voltage 28KV and current 8mA, and the raw data of stress results were obtained directly from the integrated software. The detailed information has been added in Line 98-100 on Page 3.

12. The polished and no polished samples show the same performance during tensile testing. No changes are observed.

Reply: The thickness of laser polished layer was about 60 μm, while the thickness of standard tensile samples is 3 mm (considered as bulk material). Therefore, there is no significant change of the standard tensile properties after laser polishing. Nonstandard micro-tensile specimens for thin laser-polished layer will be considered in future work.

13. English grammar requires revision throughout the manuscript in order to improve readibility.

Reply: We have carefully modified the grammar errors throughout the manuscript, as you can see this revision.

Reviewer 3 Report

This work addresses the surface polishing of Ti64 alloy by SLM. It shows interesting results and discussion and it can be considered for publication after addressing the following comments:

-          Please explain what “as-received” means? The SLM-processed sample before polishing? Please modify the term.

-          Considering Fig.2, why you mentioned alpha was formed for as-received sample? Fig. 2 (e) does not show the morphology of alpha phase. Even if it is correct, how it can be formed after high cooling rate? Please explain further. This explanation is very important as you used it for supporting the mechanical and wear properties.

-          Have you measured oxygen levels before and after polishing? The hardness improvement can be also related to oxygen pick-up after polishing. Please explain.

-          Have you considered grain size effect on the variation of mechanical and wear properties? Please explain.

-          Is there any relation between hardness and wear properties through Archard equation? Please also address wear mechanisms as well. Please see and use this work in this regard: Materials & Design 111 (2016) 592-599.

-          Why Ti-6Al-4V was chosen for your study? I don’t see any information on the importance of titanium alloys in your Introduction. Please extend it and address some of the significances of various Ti alloys including Ti64.  Please see and use these work:  Materials Science and Engineering: A 733 (2018) 80-86.

Author Response

We would like to thank the Reviewers’ invaluable comments. We have carefully considered all the comments and made corresponding modifications in the revised manuscript accordingly as listed below. All the modifications has been done are highlighted with in BLUE colour in the revised manuscript.

Note: In the authors’ reply to the reviewers, all paper numbers, figure numbers, and references are refer to the revised manuscript.

Reviewers' comments:

Reviewer #3:

This work addresses the surface polishing of Ti64 alloy by SLM. It shows interesting results and discussion and it can be considered for publication after addressing the following comments:

1.   Please explain what “as-received” means? The SLM-processed sample before polishing? Please modify the term.

Reply: “As-received” means the SLM-processed samples before laser polishing in this paper, and the similar descriptions can be found in other articles. e.g. Materials & Design, 2015, 78: 19-24; Surface Review and Letters, 2016, 23(04): 1630003; Scientific reports, 2016, 6: 28913; Optics and Lasers in Engineering, 2017, 93: 171-177; Engineering, 2018.

2.     Considering Fig.2, why you mentioned alpha was formed for as-received sample? Fig. 2 (e) does not show the morphology of alpha phase. Even if it is correct, how it can be formed after high cooling rate? Please explain further. This explanation is very important as you used it for supporting the mechanical and wear properties.

Reply: We carefully re-examined the Fig. 2. The as-received SLM samples were annealed at 1003 K (730 ℃), and the stable alpha phase and beta phases were obtained as shown in Fig.2 (e). The results are consistent with others’ work (Metallurgical and materials transactions A, 2011, 42(10): 3190-3199).

3.     Have you measured oxygen levels before and after polishing? The hardness improvement can be also related to oxygen pick-up after polishing. Please explain.

Reply: We added an EDS measurement for oxygen levels, and we found the oxygen levels were no change before and after polishing. The detailed information has been added in Fig. 6 on Page 8 and Line 212-214 on Page 6 and 7. The results are consistent with others’ work Materials & Design 111 (2016) 592-599.

4.     Have you considered grain size effect on the variation of mechanical and wear properties? Please explain.

Reply: We have carefully re-examined the effect of grain size on mechanical performance in this work. The rapid re-melting and solidification process changes the grain size at polishing layer, which improves the wear resistance of the surface. The similar results can be seen in others’ work “Tian, Y. T.; Gora, W. S.; Cabo, A. P.; Parimi, L. L.; Hand, I. D. P.; Tammas-Williams, S.; Prangnell, P. B., Material interactions in laser polishing powder bed additive manufactured Ti6Al4V components. Addit Manuf 2018, 20, 11-22”. The detailed information has been added in Line 179-181 on Page 5.

5.     Is there any relation between hardness and wear properties through Archard equation? Please also address wear mechanisms as well. Please see and use this work in this regard: Materials & Design 111 (2016) 592-599.

Reply: Thank you for your advice. On basis of the reference as “Materials & Design 111 (2016) 592-599”, the analysis of relationship between microhardness and wear properties was supplemented in the manuscript. The detailed information has been added in Line 219-223 on Page 7.

6.     Why Ti-6Al-4V was chosen for your study? I don’t see any information on the importance of titanium alloys in your Introduction. Please extend it and address some of the significances of various Ti alloys including Ti64. Please see and use these work:  Materials Science and Engineering: A 733 (2018) 80-86.

Reply: Thank you for your advice. On basis of the reference as “Materials Science and Engineering: A 733 (2018) 80-86”, the importance of titanium alloys was supplemented in the Introduction. The detailed information has been added in Line 38-41 on Page 1.
